# Generating High-Quality Explanations for Navigation in Partially-Revealed Environments

**Gregory J. Stein**
Department of Computer Science
George Mason University, Fairfax, VA
gjstein@gmu.edu

## Abstract

We present an approach for generating natural language explanations of high-level behavior of autonomous agents navigating in partially-revealed environments. Our counterfactual explanations communicate changes to interpratable statistics of the belief (e.g., the likelihood an exploratory action will reach the unseen goal) that are estimated from visual input via a deep neural network and used (via a Bellman equation variant) to inform planning far into the future. Additionally, our novel training procedure mimics explanation generation, allowing us to use planning performance as an objective measure of explanation quality. Simulated experiments validate that our explanations are both high-quality and can be used in interventions to directly correct bad behavior; agents trained via our training-by-explaining procedure achieve 9.1% lower average cost than a non-learned baseline (12.7% after interventions) in environments derived from real-world floor plans.

If we are to welcome robots into our homes and trust them to make decisions on their own, they should be able to clearly explain both *what they are doing* and *why they are doing it* in terms that both expert and non-expert users can understand. As learning (particularly deep learning) is increasingly used to inform decisions, it is equally important to ensure that data-driven systems are easy to inspect and audit both during development and after deployment [33, 6]. The field of *explanatory artificial intelligence* has developed significantly in recent years to address this need in the hopes of improving reliability and combating problematic data biases that can go unnoticed when decision-making is opaque. Myriad explainability tools aim to probe deep neural networks trained via supervised learning, often with a focus on understanding *perceptual tasks*, like detecting objects in images [24, 30, 28, 31, 38]. Yet in many real-world navigation and home care scenarios, robots are expected to make effective decisions even when their knowledge of the world is incomplete or stale; in this case, planning effectively requires making predictions about unseen space and about the potential impact of actions far into the future [39, 29, 3], complicating efforts to apply existing tools to explain robot behavior in this setting. In this work, we focus on the challenging task of *navigation in partially-revealed environments*, in which reaching an unseen goal in minimum expected time may require recognizing subtle visual cues—e.g., that green markings signal a route likely to reach the goal—and incorporating multiple such observations when deciding where to explore next.

Designing an agent that both achieves state-of-the-art performance *and* can meaningfully explain its actions has so far proven out of reach for existing approaches to planning under uncertainty. Symbolic abstraction—in which actions correspond to interaction with *symbols* (e.g., objects) or movement through the environment—is an effective tool for planning in fully-known environments and yields plans that are *interpretable-by-design*, since symbolic task plans are human-understandable by construction [25, 15]. Yet when planning under uncertainty, it is not always obvious how to abstract knowledge without problematically degrading performance, particularly when the agent perceives the world via high-dimensional sensor observations, like images. Owing to challenges associated with data and computation, many state-of-the-art approaches to planning under uncertainty *learn* the

35th Conference on Neural Information Processing Systems (NeurIPS 2021).

relationship between raw sensor observations and good behavior. Deep reinforcement learning has proven incredibly effective (if somewhat brittle [7]) in this regard [41, 36, 1, 39]. However, many such strategies are opaque by their nature and are neither well-suited to explaining themselves nor compatible with most existing tools for post-hoc explanation [12, 25].

Clearly a shift in representation is necessary: we seek an agent for which decisions are expressed in terms of symbols (so that planning is interpretable) yet can also make use of learning to ease the onerous computational and data requirements of planning under uncertainty. It is a key insight of this work that the *learning over subgoals* planning paradigm of Stein et al. [29], which blends symbolic and data-driven planning, is well-suited to meet this need. In [29], symbolic actions each correspond to revealing space beyond *frontiers*, boundaries between free and unknown space, while learning is used to estimate compact statistics of unseen space that capture the *impact* of those actions—e.g., the likelihood an exploratory action will succeed in reaching the goal. We leverage this representation to develop a new approach to generate *counterfactual explanations* that communicate what changes to these estimated statistics would result in a desired change in behavior.

In this work, we introduce a novel approach to navigation in partially-revealed environments by which an agent can achieve state-of-the-art performance and also generate *high quality* human-understandable explanations of its behavior, despite relying on learning (from visual observations) to inform decision making. Specifically, our contributions address the following three questions:

1. **What criteria must a planner satisfy if it is to be useful for generating explanations in partially revealed environments?** [Sec. 2] Our criteria (motivated by those of Lipton [15]) define what it means for planning to be *interpretable-by-design* for our problem setting.
2. **How can we generate explanations of the agent's high-level behavior?** [Sec. 3] We show that the subgoal-based planner of Stein et al. [29] meets our conditions for *interpretable* planning under uncertainty and, leveraging this representation, we introduce a procedure to generate *counterfactual explanations*. Explanations outline what changes to the agent's belief would result in a change in behavior, expressed in terms of estimated interpretable statistics of unknown space: e.g., the likelihood an exploratory action will reach an unseen goal.
3. **How can we validate that our explanations are *high quality*?** [Sec. 4] We introduce a novel training process in which training data is generated via the same procedure as our explanations. This allows us to *use planning performance as an objective measure of explanation quality*.

We evaluate the performance of our approach in two different simulated environments in which an agent is tasked to navigate to an unseen point goal in minimum expected time. We generate explanations in two different simulated environment classes and show that the explanations qualitatively match human intuition for how to correct the agent's poor behavior in these domains. Our quantitative results (Sec. 5) focus on the ability of our explanations to facilitate improving the agent's behavior and show that our joint training-by-explaining procedure demonstrates state-of-the-art performance at test time, validating that our explanations are accurate and informative and demonstrating that an agent can be both explainable and performant simultaneously in this application domain.

# 1 Preliminaries: planning in partially-revealed environments

A common way to model planning under uncertainty is as a Partially Observable Markov Decision Process (POMDP) [9, 16, 21]. In this model, the agent's behavior at time $t$ is a function of its *belief* $b_t$, a distribution over possible states of the world, which the agent updates as it receives new observations through interaction with the world. In a navigation context, the agent's objective is to minimize expected cost (travel distance) to reach an unseen goal. Belief-space planning in general is incredibly computationally intensive. Planning (typically via the *Bellman equation* [21]) requires that the agent iteratively envision how it might take action, gain new knowledge, update its belief, and then take additional actions. In the worst case, the tree of possible actions and subsequent outcomes stemming from the current belief state grows exponentially with planning horizon; planning far into the future is fraught with challenges. In this work, we study a particular class of POMDP recently coined as a *Locally* Observable Markov Decision Process [18], in which the agent's local perception is assumed to be *perfect* though limited by range and occlusions. This is a common problem setting for agents equipped with either depth sensors or LIDAR and considerably simplifies planning through *observed space*—space the agent has seen and is therefore assumed to know with perfect precision. At every timestep, the agent receives its location, an occlusion-limited local occupancy map (as if

---

Code available at `https://github.com/RAIL-group/xai-nav-under-uncertainty-neurips2021`

from a laser range finder), and an egocentric panoramic image of its surroundings and tasked to reach a goal, specified as a world-space coordinate, in minimum time. Even with perfect local perception, planning that requires the agent to reason about unseen space is still incredibly complex.

## 2 Interpretable-by-design planning in partially-revealed environments

**Criterion for interpretable planning** There are myriad choices of representations suitable for easing the challenges of planning under uncertainty—as evidenced by our earlier discussion—yet we seek in particular strategies that are said to be *interpretable*, a precondition for generating explanations [6]. A number of recent works consider what is implied by *interpretability* and propose taxonomies and criteria by which we can understand *how interpretable* a model is [6, 15, 25, 33]. Yet these works are broad in scope, requiring that we contextualize their criteria for our problem of interest. We propose the following criteria for evaluating the *interpretability* of a model that relies on learning to plan under uncertainty, motivated by those of Lipton [15], which will later serve as desiderata:

1. **Decisions available to the agent should reflect how a human might describe their own decision-making given the same information.** Low-level actions like *motion primitives* are not typically conducive for interpreting the behavior of an agent in an environment that requires thousands of steps to reach the goal. For long-horizon planning, asking *Why did the agent decide to turn* $5°$ *versus* $10°$ *degrees left?* is unlikely to yield useful explanations of the agent's decision-making process. *High-level actions* instead define temporally extended behaviors (e.g., entering a classroom or following a hallway) and often correspond to human-understandable symbols, with which planning can better match human intuition. This condition is a prerequisite to Lipton's idea of *decomposability*, that each step of planning be intuitively understandable, and is consistent with a significant body of work leveraging symbolic abstraction [5, 10].

2. **High-level planning should rely on a model and explicitly consider the impact of the agent's actions into the future.** Model-free planning strategies for planning under uncertainty, like the impressive MERLIN agent [39], directly predict the goodness of an action, *implicitly* estimating the impact that action will have on the agent's belief and thereby complicating efforts to understand the agent's behavior. Yet in many planning scenarios, humans seem to explain their decisions by *explicitly* enumerating the possible outcomes of actions that enter unknown space: e.g., whether a hallway will lead to a faraway goal. If we are to understand *why* an agent chose one action over another, its decision-making should match this structure; for planning under uncertainty to be *interpretable* and to be useful for generating explanations, the planner should (like a human) *explicitly* consider the different ways in which executing an action might accrue cost and update the belief. This condition is consistent with Lipton's *algorithmic transparency*.

3. **The impact of actions, particularly those that enter unknown space, should be expressible via a small number of human-understandable quantities.** That the plan be *compact* is important for human understanding. Yet the computational and memory requirements of belief-space planning for large long-horizon tasks are prohibitive even for a computer, and such planning fails to meet the interpretability criterion Lipton refers to as *simulatability*. The model employed by the agent must therefore dramatically simplify the belief and the process of imagining the future.

**Subgoal-based abstractions for interpretable navigation under uncertainty** It is a key observation of this work that the *learning over subgoals* planning paradigm of Stein et al. [29] satisfies our criteria for interpretable-by-design goal-directed navigation under uncertainty; we provide an overview of this approach here. Under this abstraction, *subgoals* each correspond to a high-level action to reveal unknown space by exploring beyond a *frontier*, a boundary between free and unseen space. The high-level outcome of each action is binary: an action either *succeeds*, if the agent discovers a route to the unseen goal, or *fails*, if the region is a dead-end, prompting the agent to turn back. The expected cost of each action $a_t$ (corresponding to subgoal $s_t$) depends on its likelihood of success $P_S(a_t)$ and the expected costs associated with success $R_S(a_t)$ and failure (exploration) $R_E(a_t)$. These terms (collectively, the *subgoal properties* $\boldsymbol{\sigma}$) implicitly[1] depend on the belief $b_t$ and (intractable to compute directly) are estimated via learning. The costs of navigating between subgoals involves travel through observed space and can be computed via Dijkstra's algorithm: $D(b_t, a_t)$. The expected cost of executing high-level action $a_t$ can be expressed via an approximation of the Bellman Equation

---

[1]We note that Stein et al. [29] make the dependency of the subgoal properties $P_S$, $R_S$, and $R_E$ on the belief $b_t$ *explicit*. Eq. (1) reflects our simplified notation.

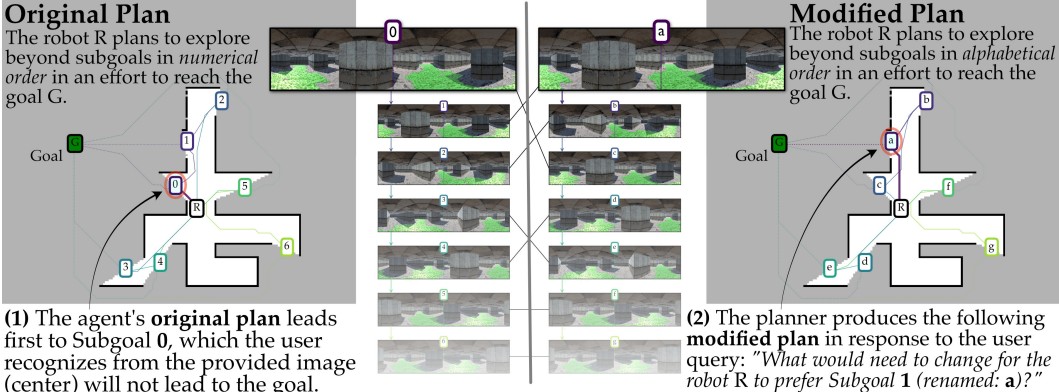

**Original Plan**
The robot R plans to explore beyond subgoals in *numerical order* in an effort to reach the goal G.

Goal

**(1)** The agent's **original plan** leads first to Subgoal **0**, which the user recognizes from the provided image (center) will not lead to the goal.

**Modified Plan**
The robot R plans to explore beyond subgoals in *alphabetical order* in an effort to reach the goal G.

Goal

**(2)** The planner produces the following **modified plan** in response to the user query: *"What would need to change for the robot R to prefer Subgoal 1 (renamed: a)?"*

**(3)** To change its behavior, the agent changes its belief about unseen space. Having computed these changes, the agent automatically generates the following **counterfactual explanation**:

> I would have preferred Subgoal **1** (renamed to **a** in the modified plan) if I instead believed Subgoal **0** (**c**) were significantly less likely to lead to the goal (36% down from 98%) and that the cost of getting to the goal through Subgoal **0** (**c**) were slightly increased (6.0 meters, up from 5.7 meters); and believed that Subgoal **1** (**a**) were slightly more likely to lead to the goal (94%, up from 86%) and that the cost of getting to the goal through Subgoal **1** (**a**) were slightly decreased (5.8 meters, down from 6.0 meters).

Figure 1: **Explaining behavior for navigation under uncertainty** The generated explanation is consistent with the user's expectations: to change its behavior, the robot must change its belief to reflect that Subgoal 0 is unlikely to lead to the goal and Subgoal 1 has a higher likelihood.

written in terms of the high-level actions and their binary outcomes:

$$Q(b_t, a_t) = D(b_t, a_t) + P_S(a_t)R_S(a_t) + (1 - P_S(a_t))\left[R_E(a_t) + \min_{a \in \mathcal{A}(b_t) \setminus a_t} Q(\tilde{b}_{t+1}, a)\right], \quad (1)$$

where $\tilde{b}_{t+1}$ is the *approximate future belief*, which incorporates the knowledge that upon executing $a_t$ and failing to reach the goal the agent has moved (and is now at subgoal $s_t$) and that the region beyond $s_t$ is now known to be a dead end (even if we do not yet know precisely what it looks like). The map of observed space is not directly updated during planning, avoiding many of the computational challenges of updating the belief. Critically, **the agent's high-level plan defines the order in which it aims to explore the unseen regions of the environment in search of the goal** specified as a list of subgoal-actions. We refer the reader to [29] for a full derivation of Eq. (1). Planning with this model is iterative. Upon selecting a subgoal via Eq. (1), a low-level planner plots a trajectory through the known map towards the chosen subgoal and the agent executes a short motion primitive along this trajectory. At every timestep, new space may be revealed, and Eq. (1) must again used to select the subgoal with lowest expected cost. This process repeats until the agent reaches the goal, which can take thousands of steps. Since the agent is assumed to have perfect local perception, the agent's ability to reach the goal is limited only by vehicle dynamics.

Given estimates of the subgoal properties ($P_S$, $R_S$, and $R_E$) for each subgoal-action available to the agent, **high-level planning via Eq.** (1) **satisfies our criteria for interpretable planning under uncertainty.** First, the decisions available to the agent correspond to choosing between exploring different regions of space, clearly human-understandable. In addition, the impact of the agent's actions (whether it reaches the goal and the expected cost of each outcome) are easy to interpret and relatively few in number. Finally, planning is done via a model, under which the impact of each action are explicitly taken into account and can influence the agent's behavior far into the future.

**Estimating subgoal properties from images** Too difficult to compute exactly, the subgoal properties $P_S$, $R_S$, and $R_E$ needed to compute expected cost are estimated via a neural network. Consistent with the approach of [29], subgoal properties are estimated from single sensor observations; thus, the input data used to estimate the subgoal properties consists of (1) a single panoramic image, (2) the relative position of the goal in the image frame, and (3) the relative position of the subgoal in the image frame, all collected at the time the subgoal was created—i.e., the time at which the subgoal's corresponding *frontier* (a boundary between free and unknown space) was revealed. A feed-forward (non-recurrent) neural network outputs the three quantities of interest; a logistic sigmoid function is applied to produce the probability of success $P_S$, so that it is a valid probability. The neural network

| **Algorithm 1:** Generate Explanation | **Algorithm 2:** Train Subgoal Property Estimator |
|---|---|
| **Data:** $m_t, q_t, a_c, a_h, \texttt{nn\_inputs}, \theta_0$ | **Data:** $\texttt{dataset}, \theta_0$ |
| **Result:** Subgoal property changes, $\Delta\boldsymbol{\sigma}$ | **Result:** Neural Network Weights, $\theta_f$ |
| 1  $\boldsymbol{\sigma}_0 \leftarrow \text{NN}(\texttt{nn\_inputs}, \theta_0)$ | 1  $\theta \leftarrow \theta_0$ |
| 2  $\Delta Q_0 \leftarrow \Delta Q(\{m_t, q_t, \boldsymbol{\sigma}_0\}, \{a_h, a_c\})$ | 2  **foreach** $\texttt{datum} \in \texttt{dataset}$ **do** |
| 3  $\theta \leftarrow \theta_0, \Delta Q \leftarrow \Delta Q_0$ | 3     $m_t, q_t, a_o, \texttt{nn\_inputs} \leftarrow \texttt{datum}$ |
| 4  **while** $\Delta Q > 0$ **do** | 4     $a_c \leftarrow \arg\min_{a \in \mathcal{A}(m_t) \setminus a_o} Q(\{m_t, q_t, \boldsymbol{\sigma}\}, a)$ |
| 5     $\boldsymbol{\sigma} \leftarrow \text{NN}(\texttt{nn\_inputs}, \theta)$ | 5     $\boldsymbol{\sigma} \leftarrow \text{NN}(\texttt{nn\_inputs}, \theta)$ |
| 6     $\Delta Q \leftarrow \Delta Q(\{m_t, q_t, \boldsymbol{\sigma}\}, \{a_h, a_c\})$ | 6     $\Delta Q \leftarrow \Delta Q(\{m_t, q_t, \boldsymbol{\sigma}\}, \{a_o, a_c\})$    ▷ Eq. (2) |
| 7     $\boldsymbol{\alpha} \leftarrow \boldsymbol{\alpha}(\Delta Q_0, \boldsymbol{\sigma}_0, \theta_0)$ | 7     $\boldsymbol{\alpha} \leftarrow \boldsymbol{\alpha}(\Delta Q, \boldsymbol{\sigma}, \theta)$    ▷ Eq. (3) |
| 8     $\mathcal{L}_{\text{comp}} \leftarrow \Delta Q$ | 8     $\mathcal{L}_{\text{comp}} \leftarrow \sqrt{1 - \text{logsigmoid}(-\Delta Q)}$ |
| 9     $\theta \leftarrow \theta - \eta \frac{\partial \mathcal{L}_{\text{comp}}}{\partial \boldsymbol{\sigma}} \cdot \left[\nabla_\theta \boldsymbol{\sigma} \circ \mathbf{1}_{\boldsymbol{\alpha} > \boldsymbol{\alpha}_{(\text{M})}}\right]$ | 9     $\theta \leftarrow \theta - \eta \frac{\partial \mathcal{L}_{\text{comp}}}{\partial \boldsymbol{\sigma}} \cdot \left[\nabla_\theta \boldsymbol{\sigma} \circ \mathbf{1}_{\boldsymbol{\alpha} > \boldsymbol{\alpha}_{(\text{M})}}\right]$    ▷ Eq. (4) |
| |        $- \eta \nabla_\theta \mathcal{L}_{\text{supervised}} - \eta \nabla_\theta \mathcal{L}_{\text{bounds}}$ |
| 10  $\boldsymbol{\sigma}_f \leftarrow \text{NN}(\texttt{nn\_inputs}, \theta)$ | 10  **return** $\theta$ |
| 11  **return** $\boldsymbol{\sigma}_f - \boldsymbol{\sigma}_0$ | |

**Algorithms for generating explanations and training our subgoal property estimator**, shown side-by-side to emphasize their similarities, in particular lines 5–9, a key feature of our approach. The details of explanation generation (Sec. 3) and training (Sec. 4) can be found in the text.

is a convolutional-then-fully-connected encoder similar to that of [3]; not the focus of this work, the network architecture is described in detail in Appendix A.1.

## 3 Generating counterfactual explanations of high-level behavior

Explanations should answer fundamental questions about the agent's behavior and provide recourse for how to correct that behavior if necessary. Let's say that, during an audit, the agent recently chose action $a_t$, yet the human auditor may have thought that $a_h$ was a preferable option. The human may want an explanation to better understand *why did the agent select action $a_t$ over action $a_h$?* The obvious (if unhelpful) answer is that the agent believed the action it selected ($a_t$) was of lower expected cost than was $a_h$. Instead, since our agent's behavior is determined by its estimates of the *subgoal properties* ($P_S$, $R_S$, and $R_E$ for each subgoal), the explanations of its behavior should feature these properties as well. Critically, since the agent's objective is to explain a *decision* to select one action over another, the explanation must contextualize the agent's choice in relation to the decision boundary between the two actions. With this in mind, **explanations generated by our system are *counterfactual*, specifying in natural language a set of changes to the (interpretable) subgoal properties that would result in a change in behavior from one action to another.**

**Computing counterfactual explanations via gradient descent** For the planning problem defined by Eq. (1), a three element tuple consisting of (1) the observed map $m_t$, (2) the robot pose $q_t$, and (3) the list of subgoal properties $\boldsymbol{\sigma}_t \in \mathbb{R}^{3 \times N_s}$ (where $N_s$ is the number of subgoals) is a sufficient statistic for the belief $b_t$. We therefore use the notation $Q(\{m_t, q_t, \boldsymbol{\sigma}_t\}, a_t)$ to represent the expected cost of executing action $a_t$ from map state $m_t$ and position $q_t$ given subgoal properties $\boldsymbol{\sigma}_t$. While the map $m_t$ and pose $q_t$ are provided to the agent, the subgoal properties $\boldsymbol{\sigma}_t$ are estimated from sensor observations via a neural network, defined by its parameters $\theta$; changing the network parameters $\theta$ would result in a change in the subgoal properties ($\Delta\boldsymbol{\sigma}$), which could change the agent's behavior.

We use gradient descent to compute our explanations, which are defined by a set of subgoal property changes $\Delta\boldsymbol{\sigma}$ and therefore a change to the agent's belief about unknown space. The decision boundary between the two actions of interest—the agent's selected action $a_t$ and the human's query action $a_h$—exists at the zero-crossing of the difference in the expected cost of the two actions:

$$\Delta Q(\{m_t, q_t, \boldsymbol{\sigma}_t\}, \{a_h, a_t\}) \equiv Q(\{m_t, q_t, \boldsymbol{\sigma}_t\}, a_h) - Q(\{m_t, q_t, \boldsymbol{\sigma}_t\}, a_t) . \qquad (2)$$

If we think of this difference in cost as a *loss* $\mathcal{L}_{\text{comp}} = \Delta Q$, crossing the decision boundary can be achieved via gradient descent over the parameters $\theta$ of the neural network used to estimate the subgoal properties $\boldsymbol{\sigma}$, which are therefore implicitly functions of $\theta$. We run gradient descent until the $\Delta Q$ crosses zero (made easy with PyTorch [20]) indicating that the robot now prefers action $a_h$ over $a_t$. The change in subgoal properties after optimization $\Delta\boldsymbol{\sigma}$ forms the backbone of our explanation.

**Ensuring explanations are compact** Returning changes to *every* subgoal property—which number three-times the number of subgoals, places the agent has yet to explore—is too much information to present all at once if explanations are to be useful for understanding or auditing an agent's behavior

[15]. As such, we wish to determine and communicate only the *most important subgoal properties* when explaining behavior. We define the *importance* of each subgoal property as their relative contribution to a change in $\Delta Q$ after one step of gradient descent. In differential form:

$$\boldsymbol{\alpha}(\Delta Q, \boldsymbol{\sigma}, \theta) \equiv \left[ \frac{\partial \Delta Q}{\partial \boldsymbol{\sigma}} \circ \nabla_\theta \boldsymbol{\sigma} \right] \cdot \frac{\nabla_\theta \Delta Q}{\|\nabla_\theta \Delta Q\|} , \qquad \text{[Subgoal Property Importance]} \quad (3)$$

where $\circ$ is the Hadamard Product (elementwise multiplication). The subgoal property importance vector $\boldsymbol{\alpha}$ satisfies $\sum_i \alpha_i = \|\nabla_\theta \Delta Q\|$ by construction. We communicate only the top $N$ most important subgoal properties as natural language explanations, including only a handful of changes to key properties instead of returning a potentially-overwhelming amount of information to the human.

If we are to claim that the explanations are accurate despite only communicating a handful of subgoal properties to the user, we must ensure that changing the agent's behavior does not rely on changes not communicated to the user. To ensure this, we use *masked gradient descent* to compute explanations: the gradients from less important properties are multiplied by zero, so that only information from the chosen *most important* properties are used to update the network parameters $\theta$:

$$\theta_{i+1} \leftarrow \theta_i - \eta \frac{\partial \Delta Q}{\partial \boldsymbol{\sigma}} \cdot \left[ \nabla_\theta \boldsymbol{\sigma} \circ \mathbf{1}_{\boldsymbol{\alpha} > \boldsymbol{\alpha}_{(M)}} \right] , \qquad \text{[Masked Gradient Update]} \quad (4)$$

where $\eta$ is the learning rate and $\boldsymbol{\alpha}_{(n)}$ is the nth *order statistic* of $\boldsymbol{\alpha}$; $M$ is defined such that $\mathbf{1}_{\boldsymbol{\alpha} > \boldsymbol{\alpha}_{(M)}}$ is a boolean vector that is true in the entries corresponding to the top-N elements of $\alpha$ (i.e., $M = |\alpha| - N$) and false otherwise. See Alg. 1 for our complete explanation generation procedure.

Fig. 1 shows an example explanation generated using our procedure; note that the explanation only includes four subgoal properties: those deemed by our agent to be the most important. We note that all subgoal properties will change during explanation generation, even when using our masked gradient descent procedure, since the network parameters $\theta$ are shared between all subgoal properties. Even so, the explanations are still an accurate reflection of the underlying decision-making process: updating the network parameters $\theta$ until the *most important* subgoal properties change by their specified amounts would cause the change in behavior the explanation seeks to describe, even if other subgoal properties are also updated in the process.

**From subgoal property changes to natural language** After generating the subgoal property changes, we turn them into a natural language explanation using a simple rule-based grammar. Subgoals are ordered by the importance of their most important property and their selected properties are inserted into template strings and concatenated. We note that the way we have chosen to present our explanations (e.g., Fig. 1) is only one way to present the information contained within our counterfactual explanations. While in this work we focus on presenting the explanations in a way that is both complete and precise, it might be that eschewing displaying exact numbers and instead including only qualitative differences improves readability. Understanding how to best present our explanations will require further study with human participants and is an important next step to more deeply understand the effectiveness of our approach; we discuss this topic further in the *future work* section of Sec. 7 and additional details can be found in Appendix A.3.

## 4    Training by explaining: validating explanation quality

A *high-quality* counterfactual explanation is both *accurate*, a faithful account of the agent's decision-making process, and is *useful*, rich with information that could be used to audit the agent and change its behavior. While our explanations are accurate by construction (see Sec. 3), the utility of an explanation can be challenging to measure. Often the measure of the quality of an explanation includes the *eyeballing metric*; this metric serves an important purpose—particularly since explaining is for communication with humans—yet Sundararajan et al. [30] point out that this approach is problematically *subjective*. Moreover, as one of the primary functions of explanations in this context will be to help improve or correct poor behavior, it is critically important to ensure that the explanations can be useful in this capacity. The structure of our training procedure and our choice of experimental results are therefore focused on this *objective* measure of an explanation's utility: whether explanations are sufficiently information-rich to facilitate correcting undesirable behavior.

With this in mind, we train our agent via the same procedure that is used to generate explanations. This *closes the loop* between the two processes, allowing us to use the performance of the system as an objective measure of the expected utility of the explanations: if the explanations were insufficient, the

agent would be unable to learn to plan effectively from those explanations and test-time performance would be poor. During an offline data-collection phase, we rely on an *oracle*—an agent with full state observability—to specify which subgoal corresponds to the shortest path of the unseen goal, thereby serving the role of the *human auditor*. At each step, a datum is recorded, including the oracle's action $a_o$, the observed map $m_t$, the agent's pose $q_t$, and the agent's observations $o$: all the information needed to estimate the subgoal properties $\boldsymbol{\sigma}$ and, with them, the expected cost $Q$.

Training relies on the generation of one-step *pseudo-explanations* that aim to describe how the estimated subgoal properties would need to change to decrease the expected cost of the oracle's action $a_o$ with respect to a *comparison action* $a_c$, defined as the lowest-expected-cost action[2] for the current datum and network parameters $\theta$. Stochastic gradient descent is used to iteratively update the neural network properties $\theta$ so as to encourage the agent to select the action $a_o$ via minimization of $\Delta Q$ for each datum. Furthermore, so as to mirror the process of generating explanations, we also use *masked gradient descent* from Eq. (4), in which only the *most important* subgoal properties are allowed non-zero gradient during training. **Each training iteration can be thought of as taking a single step towards generating a counterfactual explanation for the selected datum**. To train, we cycle through many different data; to generate an explanation, we instead iterate using the same datum until the decision boundary is crossed. Alg. 2 includes our full training procedure; note the similarities as compared to our explanation generation process shown in Alg. 1.

**Training-specific loss terms and details** We note that the objective function being optimized during training is different from the objective function during explanation generation: a few modifications are needed to help stabilize training, handle vanishing gradients, and impose additional priors on the known bounds on various quantities. While our primary objective is still to minimize $\Delta Q$, training an agent from scratch to minimize $\Delta Q$ does not yield the behavior we want. In particular, training should prioritize samples in which the agent is near to the decision boundary, since these will have the most direct impact on overall performance; optimization should focus less on scenarios in which the agent already believes that the oracle's action is best ($\Delta Q < 0$) or where modifying behavior would require significant change ($\Delta Q \gg 0$). To achieve this prioritization, instead of directly minimizing $\Delta Q$ we define an alternative *training comparison objective* $\mathcal{L}_{\text{comp}} = \sqrt{1 - \text{logsigmoid}(-\Delta Q)}$, which rescales $\Delta Q$ to (smoothly) prioritize points near to the boundary, as desired. Though a full investigation is out of scope, we anecdotally find that this alternate objective is particularly important in the more complex environments in which perfectly imitating the oracle is not possible with limited perception. Applying a monotonic scalar function to $\Delta Q$ does not change the direction of its gradient (in $\theta$-space), and therefore using this modified objective yields *identical* explanations as does using $\Delta Q$ directly in the limit of small step size. We also add two training objectives that incorporate additional knowledge and stabilize training. We add a *supervised training objective* $\mathcal{L}_{\text{supervised}}$: a cross-entropy loss that incorporates *labeled data* (whether a subgoal leads to the goal) to account for the potentially vanishing gradient associated with the logistic sigmoid function used to compute $P_S$. We also add $\mathcal{L}_{\text{bounds}}$: a collection of hinge loss terms that penalize non-physical predictions (negative values of $R_S$ and $R_E$) and imposes heuristic bounds on the expected cost $Q$, computed via a non-learned Dijkstra's algorithm planner during data collection. Finally, we note that in our *Guided Maze* environment (shown in Fig. 1) a high value of $\mathcal{L}_{\text{supervised}}$ is sufficient to achieve good performance without our training-via-explaining process, as is mentioned for a similar environment used in [29]. So as to demonstrate the effectiveness of using our explanations during training, we reduce the weight of $\mathcal{L}_{\text{supervised}}$ by a factor of 40 for all learned planners trained in that environment.

Since each datum can have over a dozen panoramic images, we use a batch size of 1 and training for each learned planner takes roughly 12 hours on a desktop Nvidia 2060 SUPER GPU. There is considerable redundancy in the data—since many images appear in multiple datum—and so we train for *only a single epoch*, yet divide the learning rate by half every time one-eighth of the data has been consumed. Full training details can be found in Appendix A.2.

## 5 Results: simulated navigation under uncertainty

We show an annotated example explanation in our *guided maze* environment in Fig. 1; an agent trained with the poor-performing learned baseline (described below) must explain why it did not select the correct path, as indicated by a green path on the ground. The explanation matches our expectations:

---

[2]If the lowest-expected-cost action during training is already the oracle action $a_o$, the next-lowest-expected-cost action is chosen for comparison instead so that training increases the margin from the decision boundary.

| Planner | Guided Maze Environment | | University Buildings Env. | |
|---|---|---|---|---|
| | Average Dist. | % Improve. | Average Dist. | % Improve. |
| ● All Subgoal Properties | **14.94** | **23.2** | 43.52 | 6.8 |
| ● 4 Subgoal Properties | **14.94** | **23.2** | **42.46** | **9.1** |
| ● No $\mathcal{L}_{\text{comp}}$ (Learned Baseline) | 19.17 | 1.3 | 44.34 | 5.1 |
| Non-Learned Baseline | 19.45 | — | 46.70 | — |

Figure 2: **Navigation under uncertainty performance** Scatterplots each show the performance of a learned planner versus the non-learned baseline for 1,000 trials in each environment; darker color indicate higher data density. Planners trained via our training-via-explaining approach outperform those that are not, validating that our explanations are useful for correcting poor behavior.

to change its behavior, the robot would need to believe that the subgoal corresponding to the only path to the goal were more likely and that its current selection were less likely (among other small changes). While this example and the explanation in Fig. 3 *qualitatively* show that our counterfactual explanations can help to elucidate the agent's decision-making process in human-understandable terms, we additionally conduct two types of *quantitative* experiments that validate the goodness of our experiments and the utility of our approach with an emphasis on demonstrating that our explanations are sufficiently rich with information that they can support correcting poor behavior.

**Planning Performance: Training via Explaining**   As discussed, our training procedure mirrors how explanations are generated, and so we can use the performance of our agent as a measure of the overall quality of our explanations. We conduct experiments in two different simulated environments: (1) the *Guided Maze*, procedurally generated mazes in which a green path on the ground indicates the (only) route to the goal and (2) the *University Buildings* environment, topologically complex maps which are extruded from over one-hundred floor plans of university buildings augmented to include obstacles to simulate clutter or furniture and in which long passages (that a human might identify as hallways) connect faraway regions of space. Our University Buildings environment, generated with aid of data from Whiting et al. [40], is quite large compared to many existing navigation benchmarks (e.g., [27]) and its size and complexity are well-suited for studying long-horizon navigation under uncertainty. Both environments are rendered via an in-house simulation environment built with the Unity Game Engine [34]. Start and goal locations are randomly selected free-space points generated via a fixed random seed, so that all planners are given the same set of tasks. We train three models for each environment type: ● All Subgoal Properties, in which all subgoal properties are used during training; ● 4 Subgoal Properties, where the gradient is limited to select only the four most important subgoal properties; and ● No Subgoal Properties (No $\mathcal{L}_{\text{comp}}$) where only the auxiliary losses $\mathcal{L}_{\text{supervised}}$ and $\mathcal{L}_{\text{bounds}}$ are used during training. The "No $\mathcal{L}_{\text{comp}}$" planner serves as our *Learned Baseline*, since it is trained without the explanatory module. We additionally compare to a *Non-Learned Baseline*, an optimistic agent that plans (via Dijkstra's algorithm) as if all unseen space were unoccupied and therefore does not rely on subgoals to plan. For each environment, we train our agents via data collected over 1,000 traversals (from start to goal) by our non-learned baseline: at every step we record a single datum containing all the information an agent may need to make a decision and the option chosen by an oracle. We evaluate performance over 1,000 traversals in *previously-unseen* test environments for each planner to show generalization; evaluation across all planners and environments takes roughly 10 days.

Our results, included in Fig. 2, show that training via our procedure yields improved results over both our learned and non-learned baseline approaches. In the Guided Maze environments, it is clear that our approach yields learned models that seems to understand the structure of the environment, showing 23.2% improvement over the non-learned baseline. Performance on the more complicated University Buildings environment further substantiates the goodness of our explanatory process: our

agent improves upon the non-learned baseline by 6.8% when trained with all subgoal properties and by 9.1% when trained using the four most important properties at each iteration. Limiting the number of subgoal properties in the University Buildings environment seems to regularize the model; exploring this effect further will be a topic of future work. In both environments, the *Learned Baseline* underperforms the other learned planners, substantiating that our *training-via-explanation* strategy (and therefore our explanations) adds information valuable for correcting unwanted behavior.

**Quantitative Interventions via Explanations** Here we show that *individual* explanations can be used to directly update the behavior of our agent via *interventions*. Generating explanations involves updating the learned model via gradient descent (see Alg. 1) until the agent's behavior changes and the counterfactual explanation therefore describes a change to the agent's learned model. In an *intervention*, those changes are *accepted* by the auditor: the agent then permanently relies on the newly updated learned model to estimate subgoal properties and plan for the remainder of the trial. A successful intervention changes the agent's *local* behavior to match the oracle for a single time step and also results in the agent continuing to perform well until the goal is reached, as measured by its overall navigation performance. For our experiments, we select the 50 trials in the University Buildings environment in which our 4 Subgoal Properties learned planner most underperforms the non-learned baseline

| Planner | Avg. Dist | Succeed |
|---|---|---|
| Non-Learned Baseline | 85.05 | — |
| No Intervention (Learned) | 124.24 | — |
| Intervened: 4 Subgoal Props. | 93.77 | 37/50 |
| Intervened: All Subgoal Props. | 92.81 | 43/50 |

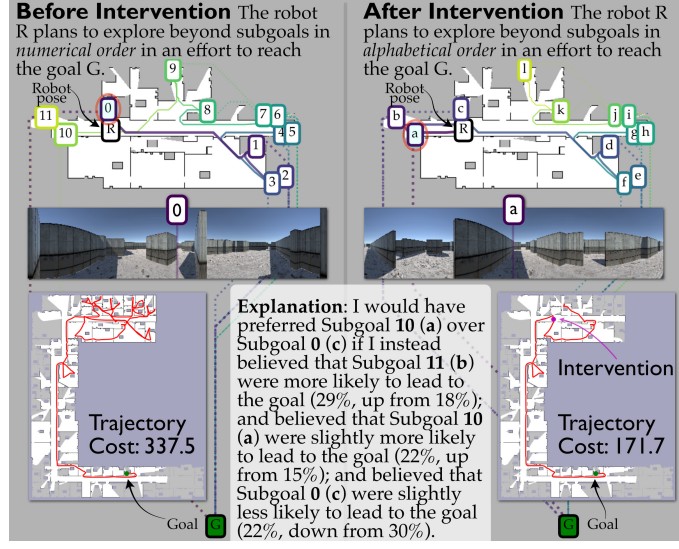

Figure 3: **Explanation-Driven Intervention Results (top) & Example (bot.)** By accepting changes proposed by explanations at key decision points, we correct poor behavior in many of the worst trials in the University Buildings environment.

and intervene at the start of the longest stretch of *bad behavior*—when the learned agent disagrees with the oracle—thus providing us with a measure of the capacity of individual explanations to improve an agent's behavior.[3] We run two sets of experiments, one in which explaining uses only 4 subgoal properties (as in training) and one where all are used.

Our intervention experiments show that individual explanations can be used to correct undesired behavior; see Fig. 3 for statistics and an example intervention in the University Buildings environment. We show an improvement of 24.5% when explaining with 4 subgoal properties and 25.3% when using all properties. The primary cause of the performance difference between the two is the increased chance of *vanishing gradients* when we limit the number of subgoals: which properties are *most important* may change as the policy is updated and so the model may be unable to cross the decision boundary, causing explaining to fail. Even when using all subgoal properties, for significant changes in behavior the neural network used to estimate subgoal properties may exhibit vanishing gradients or may propose non-physical changes (e.g., negative values for $Q$, the expected cost), which serve as grounds for rejection of the explanations. We could add additional training objectives to overcome these issues, yet these may problematically bias the explanations; exploring this relationship will be the subject of future work. Even with these occasional failures, the cost savings in our University Buildings environments from these interventions is non-trivial: 3.6% absolute cost savings when intervening using four subgoal properties and 3.7% when using all properties.

---

[3] We note that in practice, a human serves the role of the auditor, and will first review the agent's explanations before deciding whether or not a change in behavior is necessary. Our intervention experiments are designed to focus on and quantify the capacity of our explanations to facilitate these changes in behavior if requested.

# 6    Additional related work

**Explainability**    Our approach to generating counterfactual explanations is similar to a number of others that focus on the *inputs* to the learned algorithm, rather than estimates of interpretable intermediate quantities. Nearly all (like ours) rely on *gradient information* to generate counterfactuals so that explanations remain faithful to the underlying model, with a focus on either rule-based symbolic decision systems [37] and perceptual systems [38, 28, 31, 30] and with applications to natural language processing [22, 23, 8]. Our *importance* [Eq. (3)] is conceptually similar to *saliency* used in many visual explanations [31, 30], yet saliency usually *defines* the explanation, rather than being used to *guide* explaining. Recent survey and meta-analysis papers present taxonomies of the recent explosion of explainable AI tools [32, 15, 6, 25], yet none focus in particular on planning under uncertainty or its unique challenges and opportunities.

**Planning**    In addition to the black-box, model-free approaches to planning discussed above, there are a host of other techniques that impose more structure to make planning more interpretable; many are in the space of self-driving [11, 42, 26], in which the model estimates intermediate understandable quantities (e.g., where is safe to drive) *and* is trained end-to-end. These techniques are not yet well-suited for long-horizon planning or explaining high-level behavior and how to correct it. A few recent papers learn interpretable hierarchal ontologies (including so-called *neural-symbolic* planners) [14, 35, 2, 17] yet are so far primarily useful for improving performance rather than for generating explanations of high-level behavior. Recent work [19] generates post-hoc visual explanations for agents trained via deep reinforcement learning, yet focuses on short-horizon objectives.

# 7    Limitations: implications and future work

If our approach is to be broadly adopted and enable more trustworthy autonomous agents, key limitations must first be addressed. First, the model of [29] that we build upon for explaining makes strong assumptions about planning that are untrue in general, including that exploring beyond a subgoal *fully-reveals* an arm of the environment and that unknown space is *simply connected*. In environments that violate these assumptions, performance and explanation quality may be prohibitively poor. Getting good planning performance may also require that the agent "cheat" and estimate (for example) artificially-high likelihoods to bias its behavior against missing potential routes to the goal. The conflicting objectives of *high-accuracy* estimation versus predicting properties that optimize performance has potentially problematic societal implications; our system's estimates may mislead the user if they differ from their implied interpretable meaning. Second, that explanation generation can sometimes *fail* is prohibitive for widespread adoption of our approach; adding additional objectives to the explanation loss function will likely overcome this limitation (see our discussion above) yet may also problematically bias the explanations. Third, while our approach explains the agent's high-level behavior, the relationship between images and the estimated subgoal properties remains opaque, presenting challenges in the event that the human who requested the explanation is uncertain if the explanation is reasonable. There exists a body of work useful for elucidating this relation or constructing networks with interpretability in mind [4], and we look forward to future work that integrates these approaches with ours; we show preliminary results supporting the potential of these strategies in Sec. A.4. Finally, while our quantitative results are devoted mostly to demonstrating that our generated explanations are sufficiently rich with information that they can be used to improve the agent's behavior, how effectively a human can make use of these explanations to understand and subsequently correct bad behavior remains a question for future study. Though our theoretical contributions and qualitative results support the idea that our explanations are amenable for communication to humans, maximizing communication may require further tuning of their presentation. Additional work must be done to address these limitations, including additional human-in-the-loop experiments, before our approach can be trusted on an agent that coexists with humans. We additionally hope that our work will also prove useful for explaining even more complex behavior and Bradley et al. [3] recently applied the *learning over subgoals* representation to multi-stage task planning; extending our approach to support this application domain seems a promising direction for future work.

Despite these limitations, our explanation procedure represents a key step towards the design of more trustworthy robotic agents, capable of explaining their behavior even when expected to act far into the future and without full knowledge of their surroundings. Overall, we have shown that our natural language counterfactual explanations of high-level behavior for navigation under uncertainty both qualitatively meet our expectations and are quantitatively *high-quality*, that our explanation process can help to train an agent from scratch and to directly correct unwanted behavior.

## Acknowledgments and Disclosure of Funding

We thank Kevin Doherty, Leilani Gilpin, and Lucia Rafanelli for detailed feedback and thought-provoking discussions that helped to hone the ideas presented here. We also acknowledge Chris Bradley, Nick Roy, and other members of the MIT Robust Robotics group, who provided helpful feedback on an early version of this work. G. J. Stein acknowledges funding from the Department of Computer Science at George Mason University.

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
