# OpenReview forum: "Generating High-Quality Explanations for Navigation in Partially-Revealed Environments"
_NeurIPS.cc/2021/Conference — NeurIPS 2021 Poster_

### Official Review · Reviewer_74uJ · 2021-07-16

**Rating:** 5
**Confidence:** 4

**Summary:**

This paper presents a method that generates counterfactual explanations for navigation agents. The method is based on the "learning over subgoals" framework of [1], where each subgoal action indicates a plan and is formed by three properties: (i) the probability of success, (ii) the cost of success, (iii) the cost of failure. The method computes a subgoal's properties changes that can cause the agent to prefer another subgoal. These changes are computed by applying gradient descent onto the agent's model until the agent's preference switches. An explanation is then generated based on the changes, using a rule-based grammar.

The authors also propose a training-based evaluation scheme for the generated explanations. Essentially, they show that training with the gradients obtained during explanation generation leads to performance improvement.

[1] Learning over subgoals for efficient navigation of structured, unknown environments. http://proceedings.mlr.press/v87/stein18a/stein18a.pdf



**Limitations And Societal Impact:**

I suggest the authors tailor the presentation of the explanations to make them more human-friendly. Evaluating with humans is essential to keep track of the progress. The authors mention about potentially biases when evaluating with humans. However, instead avoiding human evaluation completely, the authors can also find ways to mitigate those biases through explicit instructions or obtaining a diverse pool of human judges.

**Main Review:**

**Originality**: The specific method is novel, even though it shares spirit with a lot of previous work (see related work section). To add to the list of related work, closely related ideas have recently been proposed in NLP [1, 2, 3]. The uniqueness of this work is to formulate the method in the context of partially-observable environment, and to specifically target navigation agents.

[1] Contrastive Explanations for Model Interpretability. https://arxiv.org/pdf/2103.01378.pdf

[2] Counterfactual Story Reasoning and Generation. https://aclanthology.org/2020.emnlp-main.58.pdf

[3] Back to the Future: Unsupervised Backprop-based Decoding for Counterfactual and Abductive Commonsense Reasoning. https://aclanthology.org/2020.emnlp-main.58.pdf

**Quality**:

The method for generating explanations is technically sound. It is developed on top of a well-established framework. The authors state clearly upfront the criteria that they desire from explanations and argue why their explanations satisfy these criteria.

My main concern about quality is the training-based evaluation scheme. To me, the ultimate goal of explanations is to communicate intent of agents to humans. The criteria set by the authors also strongly emphasize the human-helpfulness and human-interpretability aspects. The training-based evaluation cannot assess their aspects and thus may not be used to replace human evaluation. The scheme mainly focuses mainly on how the agent helps itself, rather than how it helps humans obtain a better understanding of its decisions. To the latter case to be true, explanations should influence decisions or knowledge of humans. This is not happening in the scheme, as the oracle (or human auditor) simply puts out a ground-truth action without consulting an explanation.

Subjectively, I find the explanation wording quite convoluted. For example, the first sentence in Figure 3  talks about why 0 is preferred over subgoal 10 but gives a reason that talks about subgoal 11. It may not be trivial to a regular user why subgoal 11 should influence the preference between subgoal 10 and subgoal 0. Human evaluation is essential to assess the interpretability of these explanations. Such a study could motivate better ways to present the explanations (e.g. using visual cues).

Similarly, it is not clear that the intervention experiment (line 321) is showing the role that the explanations are supposed to play. Here, the decisions on when to intervene are not influenced by the explanations. They are simply rule-based (i.e. choosing the first wrong actions). A more interesting that I would suggest is to let a human observe the explanations and decide when to intervene depending on whether the explanation is correct or not. For example, if the agent is not choosing a subgoal for a wrong reason (e.g. it is miscalculating its properties), the humans can give the correct reason, helping the agent alter its decision.

A concern about the results: in Figure 2 top table, in the third row, the agent should also be trained with the supervised learning loss but its performance is not much better than the non-learned baseline. I am surprised that the supervised learning loss does not help at all. The L_comp is by nature also a kind of supervised learning loss but it boosts performance dramatically. What makes the difference? Could you explain this result?

**Clarity**:

Overall, the paper is easy to follow. I really like the structure of the paper, where the clear goal is set upfront. There are a few points that need clarification (which I also request in the rebuttal):
- Description of the problem: since there are many types of navigation tasks in the literature, which one is studied in this paper? How are the navigation goals expressed (by language, image, x-y coordinates, distance to goal?)? How do you generate these goals (is there a static dataset?)? What are the training and testing conditions? Are the agent trained and tested on the same or different goals?
- Description of the environment: what is given as the state of the agent (the map, the first-person view, or both?)?What are the actions of the agent? in addition to the high-level subgoals, does the agent have low-level actions? or does it simply teleport between subgoals?
- The image of the plans are hard to understand because the annotations are not explained. What do G, R, the numbers, and the letters, the colors represent? In figure 1(2), assigning letters to subgoals while referring to them using numbers is confusing. Overall, I couldn't not understand what plans are being depicted by these figures.
- The "Quantitative Interventions via Explanations" section is missing important details. What does "accept the changes" mean mathematically? Does that mean you update the agent's model using the explanation gradient? Importantly, does the update only last during a current time step (and the agent is reset to the old parameters) or does it permanently change the agent then on? What are the changes here (from the agent's decisions to the oracle's decisions?)?

**Significance**: The paper is an attempt to extend existing explanation techniques to partially-observed navigation tasks. However, additional evaluation is required to demonstrate the effectiveness of the generated explanations in communicating the agent's intent to humans.

**Time Spent Reviewing:**

6 hours

---

> ### Author Response · Authors · 2021-08-10
> **Rebuttal comments to Reviewer 74uJ [1/2]**
>
>
> In addition to the meta-rebuttal comments above, we address more specific details here.
>
> > Originality: The specific method is novel, even though it shares spirit with a lot of previous work (see related work section). To add to the list of related work, closely related ideas have recently been proposed in NLP [1, 2, 3]. The uniqueness of this work is to formulate the method in the context of partially-observable environment, and to specifically target navigation agents.
>
> Thank you for pointing out these references; we will incorporate them into our related works section.
>
> > My main concern about quality is the training-based evaluation scheme. To me, the ultimate goal of explanations is to communicate intent of agents to humans. The criteria set by the authors also strongly emphasize the human-helpfulness and human-interpretability aspects. The training-based evaluation cannot assess their aspects and thus may not be used to replace human evaluation. The scheme mainly focuses mainly on how the agent helps itself, rather than how it helps humans obtain a better understanding of its decisions. To the latter case to be true, explanations should influence decisions or knowledge of humans. This is not happening in the scheme, as the oracle (or human auditor) simply puts out a ground-truth action without consulting an explanation.
>
> We agree that further empirical validation of our work via additional human-in-the-loop experiments is an important next step. However, particularly in the context of robotic agents that must accomplish long-time-horizon objectives, we posit that one of the primary reasons a human may want an explanation of the agent's behavior is because they wish to scrutinize and (critically) change this behavior. As such, we maintain that the human-helpfulness of the explanations in this problem setting is bounded by how useful the explanations are for improving or correcting an agent's behavior when the human deems it necessary. Given limited space, it is for this reason that we chose to focus our quantitative validation on demonstrating that our explanations have the capacity to improve the agent's behavior (across a variety of experiments) and devote the remaining space to a clear discussion of our theoretical contributions, which all reviewers point out is a strength of the paper.
>
> In an effort to make this choice clearer in the paper, we will add text to the Introduction, in the Training section (Sec. 4), and at the beginning of the Results section (Sec. 5). Additionally, as mentioned in the meta-review we will expand the Future Work section to include a discussion of the importance of future experiments involving human judges and detail what those experiments might consider.
>
> > Subjectively, I find the explanation wording quite convoluted. For example, the first sentence in Figure 3 talks about why 0 is preferred over subgoal 10 but gives a reason that talks about subgoal 11. It may not be trivial to a regular user why subgoal 11 should influence the preference between subgoal 10 and subgoal 0. Human evaluation is essential to assess the interpretability of these explanations. Such a study could motivate better ways to present the explanations (e.g. using visual cues).
>
> This is another good point related to the previous comment. We agree that the choice of grammar and the graphical representation of our explanations is an important design decision, yet one that will require a thorough investigation and experiments involving human participants. Due to space constraints, we have chosen to focus on the theoretical aspects of our explanation procedure and the capacity of our explanations to correct poor behavior. However, we agree that this could be made clearer and so we will add additional language in our Future Work section to acknowledge the utility of future experiments to establish how best to present the information in our explanations so that they may be maximally effective. We will also expand the discussion of how one might alternatively present the explanations when the grammar is discussed (at the end of Sec. 3 and in Sec. A.3) and acknowledge the utility of future experiments in this space to choose between these options.
>
> > Similarly, it is not clear that the intervention experiment (line 321) is showing the role that the explanations are supposed to play. Here, the decisions on when to intervene are not influenced by the explanations. They are simply rule-based (i.e. choosing the first wrong actions). A more interesting that I would suggest is to let a human observe the explanations and decide when to intervene depending on whether the explanation is correct or not. For example, if the agent is not choosing a subgoal for a wrong reason (e.g. it is miscalculating its properties), the humans can give the correct reason, helping the agent alter its decision.
>
> Indeed, in a real-world situation, upon questioning the behavior of the agent the human would have the opportunity to decide (in combination with the explanation) whether this behavior is reasonable and perhaps inject additional information into the system. These experiments are meant to test the limiting case of this scenario, in which the auditor *knows* that the agents behavior is sub-par and seeks to correct it via our counterfactual explanation procedure. Consistent with our discussion above, these experiments are meant to establish that our explanations are sufficiently rich with information that they can be used to correct bad behavior (and with only a single intervention), a critical capability if they are to be useful in practice. We will add language to this section of the paper to make this point clear and expand on the relationship between this scenario and how explanations will ultimately be used in practice.
>
> > A concern about the results: in Figure 2 top table, in the third row, the agent should also be trained with the supervised learning loss but its performance is not much better than the non-learned baseline. I am surprised that the supervised learning loss does not help at all. The L_comp is by nature also a kind of supervised learning loss but it boosts performance dramatically. What makes the difference? Could you explain this result?
>
> This is an insightful question, and you are correct in saying that at high-enough values the supervised loss is indeed sufficient to allow for near-perfect behavior, a point mentioned in the paper that first presents the learning over subgoals representation (our reference [24]). So as to test the utility of our explanations to improve the agent's performance, we significantly reduce the weight of the supervised loss (by a factor of 40) where (as seen in Figure 2) this leads to reduced performance of the learned baseline, thus allowing us to demonstrate the efficacy of our approach to improve training via incorporation of explanations. That this point was not mentioned in the main body text was an oversight and we will be sure to add a discussion of this point in Sec. 4 (and expand Sec. A2) to remedy it.
>
> [Our comments continue in a subsequent reply; a page break was forced by OpenReview.]

---

> > ### Author Response · Authors · 2021-08-10
> > **Rebuttal comments to Reviewer 74uJ [2/2]**
> >
> > [Our comments continue from above]
> >
> > > Clarity:
> > > Overall, the paper is easy to follow. I really like the structure of the paper, where the clear goal is set upfront. There are a few points that need clarification (which I also request in the rebuttal):
> > > Description of the problem: since there are many types of navigation tasks in the literature, which one is studied in this paper? How are the navigation goals expressed (by language, image, x-y coordinates, distance to goal?)? How do you generate these goals (is there a static dataset?)? What are the training and testing conditions? Are the agent trained and tested on the same or different goals?
> >
> > The agent's navigation goals are provided as coordinates and the start and goal locations are randomly selected points, yet are made "static" by a fixed random seed so that all planners are given identical trials. We will update the Preliminaries (Sec. 1) to better specify the problem statement and the description of the environments (Sec. 5) [see our comments below]. We will expand the text in Sec. 5 that mentions that the training and testing environments (including the goals) are distinct.
> >
> > > Description of the environment: what is given as the state of the agent (the map, the first-person view, or both?)?What are the actions of the agent? in addition to the high-level subgoals, does the agent have low-level actions? or does it simply teleport between subgoals?
> >
> > Reviewer 5zEg had similar questions, and we will include an expanded discussion of the agent's perceptual model and of the planning process as part of our description of the problem setting in the Preliminaries section (Sec. 1). In summary, the agent receives a local occupancy grid (as if from a laser range finder) and an egocentric panoramic image. A non-learned low-level motion planner is responsible for making incremental progress through known space to the selected subgoal. Planning is iterative: both the high-level planner (which selects the lowest-cost subgoal) and the low-level planner are run at every timestep until the agent reaches the goal.
> >
> > > The image of the plans are hard to understand because the annotations are not explained. What do G, R, the numbers, and the letters, the colors represent? In figure 1(2), assigning letters to subgoals while referring to them using numbers is confusing. Overall, I couldn't not understand what plans are being depicted by these figures.
> >
> > We agree that the presentation in the figures could have been better and we have remade the figures (and caption) to address these ambiguities. In our updated figures 1 and 3, we clarify via annotations that "The robot 'R' plans to explore beyond the subgoals in numerical (alphabetical) order in an effort to reach the unseen goal 'G'" before (after) the plan modifications specified by the explanation; we hope this clarifies the meaning of the symbols as well as why the subgoal labels were changed between the different images. The addition of these annotations, as well as a few more labels, we believe will make the figures (and therefore our results) easier to understand. We will also updated the captions and explanations to refer to the subgoals by both their number and letter, so that the subgoals can be easily identified in both locations.
> >
> > > The "Quantitative Interventions via Explanations" section is missing important details. What does "accept the changes" mean mathematically? Does that mean you update the agent's model using the explanation gradient? Importantly, does the update only last during a current time step (and the agent is reset to the old parameters) or does it permanently change the agent then on? What are the changes here (from the agent's decisions to the oracle's decisions?)?
> >
> > Another reviewer (qvY1) had this question as well; we will add to this section of the text to clarify the process. The counterfactual explanation depends on a change to the agent's learned model. In the interventions, those changes are "accepted" and the agent then permanently relies on the newly updated learned model (post-explanation-generation) to estimate the subgoal properties and plan until the goal is reached. The experiment shows that, not only can we change the agent's behavior to match the oracle for a single time step, but that the agent continues to perform well for the remainder of the trial.
> >
> > > Significance: The paper is an attempt to extend existing explanation techniques to partially-observed navigation tasks. However, additional evaluation is required to demonstrate the effectiveness of the generated explanations in communicating the agent's intent to humans.
> >
> > Please see our comments above and our discussion in the meta-rebuttal, which addresses this point in relation to your previous comments and suggestions.
> >
> > > Limitations And Societal Impact: I suggest the authors tailor the presentation of the explanations to make them more human-friendly. Evaluating with humans is essential to keep track of the progress. The authors mention about potentially biases when evaluating with humans. However, instead avoiding human evaluation completely, the authors can also find ways to mitigate those biases through explicit instructions or obtaining a diverse pool of human judges.
> >
> > We agree that human trials are an important next step before our explanatory process be suitable for mission-critical applications, and will add additional language to the throughout the paper (see our discussion above) and in our "Limitations" section (Sec. 7) to address this point further. Yet given the importance of thoroughness in human trials and the space limitations for this paper, we have focused instead on the more theoretical aspects of our procedure as well as another important metric for establishing their utility (as discussed above) within this challenging application domain. We refer to our comments above for additional discussion of this point.

---

> > > ### Author Response · Authors · 2021-08-26
> > > **Author Followup to Reviewer 74uJ**
> > >
> > > Reviewer 74uJ: We would just like to follow up to see if you still had any additional questions after our comments made during the author response period. As we have entered the final week of the reviewer discussion period, we are available to further clarify any of the points we made during our direct reply to your comments or in our meta-response above. In particular, we highlight that the review comments have helped us to clarify a few technical details and to hone the scope and narrative of our paper. Thank you again for your constructive feedback.

---

> > > > ### Comment · Reviewer_74uJ · 2021-08-28
> > > > **Thank for the response. Some follow up questions!**
> > > >
> > > > Thank you for the detailed response. I would like to ask additional questions to better understand your work. I have the answers for myself and want to compare with yours.
> > > >
> > > > - According my read of your paper and response, a major strength of your method is to enable the human to correct behavior of the agent. However, I can accomplish that with any imitation learning algorithm (e.g. by minimizing some loss with respect to the ground truth action). What are the advantages of your method compared to standard imitation learning methods, other than the intepretable input features (the sigma)?
> > > >
> > > > - Regarding to Figure 2, as I understand, the third row of the table features an agent that is not trained with L_comp but is trained with L_supervised. However, the weight of L_supervised is reduced by 40 times (which is equivalent to setting a very small learning rate?). What is the significance of this baseline? Are you training it with the same number of iterations as the other agents are trained? I can't see how this baseline is significantly different from the non-learned baseline.

---

> > > > > ### Author Response · Authors · 2021-08-29
> > > > > **Clarifications for Reviewer 74uJ**
> > > > >
> > > > > Thank you for reaching out with additional clarifying questions; we address these below.
> > > > >
> > > > > **Similarities to imitation learning**
> > > > >
> > > > > This is a good point and we note that it may be helpful to think of our training procedure *as a type of imitation learning* for subgoal-based planning in this domain, and so contrasting the two is perhaps not meaningful. We note that indeed the primary theoretical advantage of our particular choice of training procedure over other potential strategies to train an agent in this domain is the deep connection with our process for generating explanations, which allows us to establish a connection between the performance of the agent and the expected utility of the explanations to correct the agent's behavior. We will update the paper to acknowledge the connection between our training procedure and training via imitation.
> > > > >
> > > > > We additionally point the reviewer to the intervention experiments, which empirically show that the same procedure can be used to improve the agent's behavior at test-time with only a single "training example" (intervention), resulting in better average behavior overall and highlighting the utility of our choice of model. Despite the scale and complexity of our University Building environments, we demonstrate that intervening at a single time does not problematically degrade performance afterwards, when the agent must use its updated learned model to reach the unseen goal.
> > > > >
> > > > > **Clarifying details regarding the baseline planning strategies**
> > > > >
> > > > > These are good questions and we will be sure to clarify these points in the text.
> > > > >
> > > > > First, L_supervised is a regularization term: it appears in *all learned planners* and the weight of L_supervised is constant across all the learned planners of a particular environment. L_supervised is *uniformly* smaller for all learned planners in the Guided Maze environment so as to make those experiments a more meaningful demonstration of the utility of training with L_comp. For all experiments, other than the fact that L_comp is multiplied by zero for the learned baseline, the parameters and training details of the learned baseline are identical to those of the other learned policies. Our results in the Guided Maze environment in particular show that our learned agents trained with L_comp have significantly-improved performance over the learned baseline, a necessary point of comparison to show that L_comp was of critical importance for success in our Guided Maze experiments.
> > > > >
> > > > > Second, the non-learned baseline plans via a different strategy that does not consider subgoals when planning. The non-learned agent instead plans via Dijkstra's algorithm over an occupancy grid in which all unseen space is assumed to be unoccupied (a common planning strategy for our agent's perceptual model). Even a randomly-initialized neural network will produce subgoal properties that, used to plan via Eq. 1, yield potentially different behavior from the non-learned baseline. We also refer the reviewer to the third plot (colored in blue) on the left side of Figure 2, showing that despite the similar statistical performance of the Learned Baseline and Non-Learned Baseline, the performance of each can differ significantly on a trial-by-trial basis.
> > > > >
> > > > > We will be sure to update the paper to clarify these points and to emphasize the differences between the planners.

---

> > > > > > ### Comment · Reviewer_74uJ · 2021-08-30
> > > > > > **Thank you very much. I appreciate the response!**
> > > > > >
> > > > > > **Clarifying details regarding the baseline planning strategies**: The answer makes a lot of sense. It has addressed my concern.
> > > > > >
> > > > > > **Similarities to imitation learning**: I was also thinking of your method as an instance of imitation learning. I was not asking you to contrast the method with imitation learning. I was asking if there are **additional** advantages of your method compared to a standard imitation learning method. My point is that I could use any imitation learning method to alter the agent's behavior. The fact that the explanation helps altering the agent's behavior is not surprising to me because it is the gradient w.r.t. the input features. So for the audience who would be reading the paper, the important question that you may need to address is: why should I choose your method over other imitation learning methods?
> > > > > >
> > > > > > In my humble view, there are two possible ways to answer this question:
> > > > > > - Focusing on the ability of the explanation to improve the agent performance (which you are doing). If you insisted on this path, I'd like to see comparison with other imitation learning methods or at least variants of the imitation loss. The fact that your method can improve the agent performance with a single example seems impressive but it is not too meaningful without comparing with a baseline that could achieve the same thing (but less efficiently).
> > > > > > - Focusing on interpretability. This is more exciting to me. Your method not only allows a human to the agent's behavior but it could also help them decide **when** to do it, because they could read and understand the explanations. The latter aspect would separate your method from imitation learning: those methods can only help the human change agent's behavior but cannot help them decide when to do it. If you went on this path, evaluation with real humans is crucial to demonstrate the effectiveness of the method.

---

> > > > > > > ### Author Response · Authors · 2021-08-30
> > > > > > > **Followup for Reviewer 74uJ**
> > > > > > >
> > > > > > > We are glad that our comments helped your understanding and appreciate your continued interest in the paper.
> > > > > > >
> > > > > > > We agree that a focus on interpretability (your second bullet above) is an exciting direction, and indeed this is the central focus of our work. We also agree that human-centric experiments will need to be conducted to better understand the effectiveness of our method before it is deployed in mission-critical, human-in-the-loop scenarios. Yet we believe that the work we have presented here already represents a significant advance in how to think about explainable planning in this challenging application domain, even in the absence of human-in-the-loop experiments, a point we expand upon below.
> > > > > > >
> > > > > > > First, regarding specifically the utility of the intervention experiments: the work we have presented here is a *prerequisite* to additional, comprehensive human-in-the-loop trials. Supported by the theoretical contributions and qualitative results presented in the remainder of the paper, our intervention experiments provide an "upper baseline" against which human performance can be compared. It is only therefore *because* of the success of our intervention experiments that we know it is possible for humans to improve the agent's overall behavior in this setting in a way that is consistent with explanation generation. This potential is further supported by our qualitative results, including the figures showing the explanations themselves and the additional proof-of-concept pixelwise saliency results we conducted in response to a comment from Reviewer 5zEg.
> > > > > > >
> > > > > > > To summarize some of our previous comments above: *our work is an important prerequisite to human trials and therefore lays the foundation for additional advances in this space.* The fact that it is possible to correct an agent's behavior in a way that is consistent with explanation generation for long-horizon planning under uncertainty is a novel contribution made possible for the first time by the theoretical insights we present in this work. Moreover, space limitations would make it challenging to include such rigorous, comprehensive human trials without detracting from our theoretical contributions and experimental results or without hurting clarity, which all reviewers highlight as a strength of the paper.
> > > > > > >
> > > > > > > We do acknowledge that we could be clearer about these important points in the paper and, consistent with our comments above, we will add language throughout (especially in the Limitations and Future Work sections) to clarify them.

---

### Official Review · Reviewer_5zEg · 2021-07-16

**Rating:** 7
**Confidence:** 4

**Summary:**

- This paper discusses the challenges of providing high-quality explanations of agent navigation decision making in partially-observable environments.  PointGoal in particular is studied.
- The authors proposed a transparency-by-design method based on subgoal-based actions.  The agent selects a subgoal in the form of an area to explore next and selection is based on simple criteria.
- Given an alternative subgoal, an explanation can be generated for why the agent selected its chosen subgoal instead of alternative subgoal.
- The efficacy of the explanations are validated by showing the performance of the agent.  Since the explanations are directly part of the agent's action selection process, showing that the agent performs well also ensures that the explanations are high-quality.  The agent performs well and training via explanation helps.
- Finally, the authors show that the agent can be provided interventions to improve performance


**Limitations And Societal Impact:**

Yes

**Main Review:**

Post Rebuttal
-----------------

I thank the authors for their response. I have increased my score to a 7 to indicate that I think the paper should be presented at NeurIPS.

One comment: SPL is still very useful when success is nearly 100% as then it functions as a normalized version of Average Dist. In my opinion, SPL should be reported in addition to Average Dist. as this is a metric that readers from the visual navigation community will be highly familiar with, thereby improving the readability of the paper.

Original Review
---------------------
- Strengths
    - This paper approaches a difficult topic, explanations for visual navigation agents, and provides useful and insightful discussion on the topic
    - The agent action selection procedure and explanation procedure are both well-thought-out
    - The paper is very well-written and easy to follow
    - The interventions work well
- Weaknesses
    - Are the explanations better than "$$a_t$$ was selected because the agent believed it was of lowest cost" in practice?  They provided more detail, but the explanation are based on estimations from a neural network, not wrt inputs themselves.  In other words, while the actual action selection mechanism may be interpretable and transparent, the inputs to that mechanism are not.  I do not see why a human auditor would be able to determine why (using the example in Fig 1) Subgoal 1 has a 86% chance of success instead of a 94% chance.
        - The authors do note this issue and note that work on NN's interpretability may be able to help, but I am unsure if those methods are able to generate explanations in this situation since the NN is tasked with both modeling the low-level controllers ability to "explore beyond a frontier" and the input to it is more indirect than in say classification -- the network is given an input image and tasked with using that to infer properties about the rest of the environment to then estimate whether or the agent will succeed.
        - A proof of concept experiment that existing CNN interpretability techniques provide something reasonable here would greatly strengthen this paper.
    - Non-standard evaluation procedure and dataset.
        - The datasets used for evaluation aren't standard for PointGoal Navigation.  Ideally the datasets from Gibson or Habitat should be used as these would allow for comparison with many other methods (albeit non-explainable ones).
        - The evaluation metric of Average Dist. is also not standard nor is it explained in text.  Adding in metrics such as Success and SPL (https://arxiv.org/abs/1807.06757) would improve the paper.
        - It is never stated if there is a train/test split or if there is no generalization be tested
        - Will the Guided Maze and University Buildings datasets be released?
    - Questions:
        - How is "explore beyond a frontier" performed?  That seems very ill-defined.
- Overall
    - While I see issues with this work, I don't necessarily think they should block publication as I do believe others may find this work useful and insightful.  I look forward to discussing this work with other reviewers.


**Time Spent Reviewing:**

2

---

> ### Author Response · Authors · 2021-08-10
> **Rebuttal comments for Reviewer 5zEg**
>
>
> In addition to the meta-rebuttal comments above, we address more specific details here.
>
> > Strengths
> > This paper approaches a difficult topic, explanations for visual navigation agents, and provides useful and insightful discussion on the topic
> > The agent action selection procedure and explanation procedure are both well-thought-out
> > The paper is very well-written and easy to follow
> > The interventions work well
>
> Thank you! We are glad that you found the paper easy to follow.
>
> > Weaknesses
> > Are the explanations better than "a_t was selected because the agent believed it was of lowest cost" in practice? They provided more detail, but the explanation are based on estimations from a neural network, not wrt inputs themselves. In other words, while the actual action selection mechanism may be interpretable and transparent, the inputs to that mechanism are not. I do not see why a human auditor would be able to determine why (using the example in Fig 1) Subgoal 1 has a 86% chance of success instead of a 94% chance.
>
> We address the first point of this comment in response to the subpoints below.
>
> The second point partly motivated comments we make in the meta-rebuttal and the subsequent proposed changes to the paper. We believe that the other detail in the example highlighted in Figure 1 helps to illustrate how effective the subgoal properties can be in illuminating the choices made by the agent. In particular, the explanation shows that the agent is a bit too "optimistic" in its original predictions about the likelihood of each action to lead to the goal. The biggest change expressed in the counterfactual is the drop in likelihood of Subgoal 0 from 98% to 36%, which is easier to understand since the new (lower) value much more closely agrees with what we know to be true about the environment: that paths without green markers on the ground are not expected to lead to the goal, something the human can observe from the onboard images collected by the agent.
>
> However, to your last point---regarding the 86% to 94% increase---we agree that it may not be clear the extent to which "smaller" changes like this are useful to a human operator, whose intuition for the structure of the environments or how the agent behaves may not be so precise, and one may question the utility of including such changes in the counterfactual. It is our aim in this work to ensure that our explanations are unabridged, accurate reflections of the agent's behavior and that these explanations are maximally useful for correcting that behavior when necessary. As mentioned in the meta-rebuttal, in future work we hope to explore how to best present this information to humans: e.g., changing the number of subgoal properties we use during masking, the grammar we use to generate the language, or the graphical interface. We will update the text in both the Limitations & Future Work section (Sec. 7) and when discussing the natural language grammar (Sec. 3 and Sec. A.3) to further clarify our choices in this regard as well as to suggest potential alternatives.
>
> > The authors do note this issue and note that work on NN's interpretability may be able to help, but I am unsure if those methods are able to generate explanations in this situation since the NN is tasked with both modeling the low-level controllers ability to "explore beyond a frontier" and the input to it is more indirect than in say classification -- the network is given an input image and tasked with using that to infer properties about the rest of the environment to then estimate whether or the agent will succeed.
> > A proof of concept experiment that existing CNN interpretability techniques provide something reasonable here would greatly strengthen this paper.
>
> We agree that a proof of concept showing the potential of existing interpretability tools to inspect our trained network would be a useful addition to demonstrate the feasibility of future work in this regard. We have conducted some preliminary experiments that will be added to a new appendix A.4. As mentioned in the meta-rebuttal, the results qualitatively agree with our expectations---highlighting the most salient feature in the environment across different subgoals---which we believe supports the potential to enhance our approach via integration with other existing tools.
>
> We additionally note that the low-level planner is non-learned, and so our learned model does not need to directly reason about the local behavior of the agent. As part of our efforts to clarify the process by which the agent plans, we will add text to Sec. 1 and 2 to help clarify this point. See also our comments in the meta-rebuttal and in our reply to Reviewer 74uJ, in which we mention additional changes we will make to further clarify the problem setting and the planning process.
>
> > Non-standard evaluation procedure and dataset.
> > The datasets used for evaluation aren't standard for PointGoal Navigation. Ideally the datasets from Gibson or Habitat should be used as these would allow for comparison with many other methods (albeit non-explainable ones).
> > The evaluation metric of Average Dist. is also not standard nor is it explained in text. Adding in metrics such as Success and SPL (https://arxiv.org/abs/1807.06757) would improve the paper.
>
> These are both good comments and we agree that this could be made clearer. We note that our problem setting is somewhat different to what is often under study in the Habitat and Gibson environments, in which low-level perception is used to support both low-level and "mid-level" planning (roughly ~10 meters into the future). Our agent instead assumes perfect local perception and relies on a non-learned low-level planner to navigate through revealed space (a point we will make more clear upon revision). In addition, compared to the Habitat and Gibson environments, in which trajectories greater than 20 meters are considered long (see, e.g., [*1]), our environments are quite large and full of lengthy dead-ends, and our agent regularly requires trajectories of over 100 meters to reach the unseen goal; these larger environments allow us to study the ability of our agent to envision the impact of its actions far into the future and avoid dead-ends; we will mention this is Sec. 5. Additionally, since our success rate is nearly 100%, SPL may not be the most meaningful way to report performance. Instead, we follow the convention for POMDP planning (e.g., refs [8] and [14]) to directly report expected cost (in units of distance) for each planner, computed by averaging performance across a large number of trials and additionally supported by the scatterplots shown in Fig. 2. Finally, we will update Sec. 2 to further clarify the perceptual model and planning process of our agent and that success is limited only by vehicle dynamics (and not the performance of our learned model).
>
> [*1] DD-PPO, Wijmans et al. https://arxiv.org/pdf/1911.00357.pdf
>
> > It is never stated if there is a train/test split or if there is no generalization be tested
>
> All test-time environments are distinct from those seen at training time; we do mention this very briefly on L308, but we admit that this could be made more clear and will add some text to emphasize this point.
>
> > Will the Guided Maze and University Buildings datasets be released?
>
> Yes. The simulated environment as well as the floorplans used to generate the University Buildings maps are included in the supplementary material, and both will be included in the public code release upon publication.
>
> > Questions: How is "explore beyond a frontier" performed? That seems very ill-defined.
>
> You were not the only reviewer to ask about this point, and we will make this clearer as part of our additions to Sec. 1 and Sec. 2. At every time step, the high-level planner picks a subgoal to navigate towards (each subgoal corresponds to a single frontier) and our low-level planner plans a trajectory through known space to reach this point, along which the robot makes incremental progress. As new space is revealed, the subgoals change, learning is used to estimate new subgoal properties, and the high-level planner runs again; over time the agent will explore more and more space until the goal is reached. Both planners are therefore run at every timestep.
>
> > Overall:
> > While I see issues with this work, I don't necessarily think they should block publication as I do believe others may find this work useful and insightful. I look forward to discussing this work with other reviewers.

---

### Official Review · Reviewer_qvY1 · 2021-07-17

**Rating:** 7
**Confidence:** 3

**Summary:**

This paper presents a method for deriving counterfactual explanations of model decisions (relative to the correct decision the agent should have made). The general method is that by performing gradient descent over the model parameters until its decision changes, a set of properties is identified as important features which the agent had mistakenly used to make its decision. To ensure that a minimum number of properties is identified (only the important ones), a second pass of gradient descent is used fixing all but these properties to ensure they still result in a change in the decision. The method is evaluated by training a model using these explanations.

**Limitations And Societal Impact:**

I think the limitations of the approach were well addressed. One minor limitation I was curious about is how this approach might extend to environments that are dynamic, or action spaces which including manipulating an environment.

**Main Review:**

Originality: I am not intimately familiar with work regarding explainable systems / generated explanations, so it is difficult for me to place this in relation to prior work. However, the proposed method seems original relative to the related work described in the paper.

Quality: The method seems well-supported theoretically and in the simulated experiments. I would be interested to see performance without the supervised auxiliary loss. Also, I would have been interested to see results using human judgments, although prior work has found these problematically subjective. This is  not within the scope of this work, but it is surprising that there is not a reliable way to evaluate how useful explanations are to users of a system.

The authors describe several limitations to the approach, which I appreciated. I was somewhat confused about the space of high-level actions: it seems that (from Sections 2 and 7) taking a single high-level action would fully reveal the remaining environment accessible via taking that action. I am somewhat confused why this is the case (if I am understanding correctly), and what challenges would need to be addressed to remove this assumption. Similarly, I was curious about the topologies of the maps: do they have cycles? Multiple paths to a goal? How many high-level actions are available in the start state?

I appreciated the care put into the design of the method, including masked gradient descent.

Clarity: There were several points which I found confusing. I would have preferred a bit more formalization of the task itself and the learning setup. And a few questions below:
- What clues does the agent have in the university environment that one location might lead to the goal? In the other environment, it seems green paths would be the clue.
- The explanations explain a *mistake* the agent makes, conditioned on the correct action to take. However, they don't seem to explain the initial prediction made by the agent, conditioned on the image input. They also don't seem to explain why the changes in the subgoal properties result in the change in the predicted action. Is my understanding correct? (Section 7 seems to confirm this, but I'm not sure.)
- What does the space of subgoal properties look like? Does the agent have true access to these when navigating or is this something it is estimating?
- I was confused about how learning from explanations worked in Section 4. Is this just that upon a mistake, the gradient descent procedure is carried out, and the resulting parameters are used in the next environment rollout?
- Typo in L175: "boundary"

Significance: Some of the limitations discussed in Section 7 make me wonder about this method's applicability to new environments and action spaces; however, the general idea behind this proposed method seems like it could be extended.

**Time Spent Reviewing:**

2

---

> ### Author Response · Authors · 2021-08-10
> **Rebuttal comments for Reviewer qvY1**
>
>
> In addition to the meta-rebuttal comments above, we address more specific details here.
>
> > Originality: I am not intimately familiar with work regarding explainable systems / generated explanations, so it is difficult for me to place this in relation to prior work. However, the proposed method seems original relative to the related work described in the paper.
> > Quality: The method seems well-supported theoretically and in the simulated experiments. I would be interested to see performance without the supervised auxiliary loss. Also, I would have been interested to see results using human judgments, although prior work has found these problematically subjective. This is not within the scope of this work, but it is surprising that there is not a reliable way to evaluate how useful explanations are to users of a system.
>
> The auxiliary losses exist to help stabilize training; without these losses, training is sometimes ineffective and the performance of the cannot reliably exceed the performance of the non-learned baseline. For example, preliminary experiments (conducted during the rebuttal period) demonstrate that without these losses, the performance of the 4SG planner in the Guided Maze environment roughly matches the non-learned baseline. We will expanded Section A.2 to include this discussion, and we can provide more detailed statistics in the camera-ready version. It is also noted in Limitations (Sec. 5), that it is possible for the agent to perform well in the Guided Maze environments but with unrealistic values of the subgoal properties; the auxiliary losses help avoid this. We will expand discussion of this point.
>
> > The authors describe several limitations to the approach, which I appreciated. I was somewhat confused about the space of high-level actions: it seems that (from Sections 2 and 7) taking a single high-level action would fully reveal the remaining environment accessible via taking that action. I am somewhat confused why this is the case (if I am understanding correctly), and what challenges would need to be addressed to remove this assumption.
>
> The high-level actions are themselves an abstraction of the future behavior of the agent (first introduced as part of the learning over subgoals model in ref [24]) and are used during planning to allow the agent to more easily understand the impact of its actions. At each timestep, the agent takes a single (low-level) step towards its chosen subgoal (corresponding to a high-level action), potentially reveals new space, and then queries the high-level planner again (as described in Sec. 2) to pick another subgoal. An alternative approach imagines what unseen space might look like for the entire remainder of the environment and is a prohibitively expensive option. This model has been shown by [24] (and again in this paper) to considerably simplify planning in partially-revealed environments. As we mentioned in the meta-rebuttal, we will update Sec. 2 to elucidate how the learning over subgoals model is used to improve planning under uncertainty.
>
> > Similarly, I was curious about the topologies of the maps: do they have cycles? Multiple paths to a goal? How many high-level actions are available in the start state?
>
> The example maps that we show in the figures are good examples of what the two different environments look like in general: the Guided Maze environments have no cycles and only a single path to the goal while the University Buildings are topologically complex and will often have multiple routes to the goal. The high-level actions correspond to boundaries between free and known space and are therefore dynamic: as more space is revealed, the number of high-level actions will change. In the university buildings environment, the agent can have dozens of available actions to choose between. We will add text to Sec. 2 and Sec. 5 to clarify these points.
>
> > I appreciated the care put into the design of the method, including masked gradient descent.
> > Clarity: There were several points which I found confusing. I would have preferred a bit more formalization of the task itself and the learning setup. And a few questions below:
>
> The other reviewers also asked for some additional clarification about this. We will add text to Sections 1 and 2 as part of our overall update clarifying the robot's objective and the planning process (see our meta-rebuttal for a summary and discussion of these changes). We discuss the estimated subgoal properties in more depth in response to another of your comments below.
>
> > What clues does the agent have in the university environment that one location might lead to the goal? In the other environment, it seems green paths would be the clue.
>
> A human navigating in a similar environment would likely keep mostly to hallways while sufficiently far from the goal. Though our agent is not told what a hallway is, we expect it to learn that long corridors (that a human might identify as a hallway) connect faraway regions of space and are more likely to lead to a goal. This characteristic is visible in the images (see, for example, Fig. 3) and is a queue we expect the agent to pick up on. We will mention this feature of the environment in our discussion in Sec. 5.
>
> > The explanations explain a mistake the agent makes, conditioned on the correct action to take. However, they don't seem to explain the initial prediction made by the agent, conditioned on the image input. They also don't seem to explain why the changes in the subgoal properties result in the change in the predicted action. Is my understanding correct? (Section 7 seems to confirm this, but I'm not sure.)
>
>
> To your first point, your understanding is correct: our work primarily focuses on the ability of our explanation procedure to elucidate the high-level (long-time-horizon) behavior of the agent. It is in Sec. 7 that we acknowledge the potential utility of augmenting our explanations to show (for example) feature attributions associated with our estimated subgoal property (i.e., how the subgoal properties are estimated from raw images). Motivated by this comment and another by Reviewer 5zEg, we will add preliminary proof-of-concept experiments (described in the meta-rebuttal above) in a new Appendix A.4. Our preliminary results show that existing work on interpreting convolutional neural networks produces qualitatively reasonable results and thus illustrating the potential utility of our explanations in this regard. Additionally, Section 2 describes the condition of algorithmic transparency in the context of navigation under uncertainty: based on their own experience navigating through places they've never been, we expect users to have some intuition that if a particular avenue is unlikely to lead to the goal, one should not expend energy exploring it. We will clarify this point in the text.
>
> > What does the space of subgoal properties look like? Does the agent have true access to these when navigating or is this something it is estimating?
>
> The three properties associated for each are all scalar quantities (one probability, and two expected cost terms) estimated via a neural network, trained by our "training by explaining" procedure described in Sec. 4 with data collected via the non-learned baseline planner (mentioned in Sec. 5). Collectively, the subgoal property vector $\sigma$ is therefore a 3 x number-of-subgoals length vector (which we will mention in Sec. 3). During navigation the agent must estimate these properties for every subgoal, since it does not know their true value, which it then uses to select a subgoal to navigate towards. We appreciate that the full planning process and the agent's perceptual model (including the role played by the neural network) were somewhat unclear in the first version of this submission, and so we hope that our additional text to clarify these above (which will be added to Sec. 1 and Sec. 2) will help to avoid this confusion in the future.
>
> > I was confused about how learning from explanations worked in Section 4. Is this just that upon a mistake, the gradient descent procedure is carried out, and the resulting parameters are used in the next environment rollout?
>
> This question is similar to one asked by Reviewer 74uJ. Planning in our environments proceeds iteratively over thousands of timesteps. When we intervene, the changes proposed by the explanation are made permanent (which we refer to as ``accepting'' those proposed changes) and the agent uses the newly-updated network to plan until the end of the trial. The experiments aim to show that intervening in the agents behavior in this way has the capacity to change instantaneous performance and its impact on future performance. We will add text to Sec. 5, which (in addition to our clarifications of the planning process we will add to Sec. 1 and 2) will help clarify this point.
>
> > Typo in L175: "boundary"
>
> Fixed!
>
> > Significance: Some of the limitations discussed in Section 7 make me wonder about this method's applicability to new environments and action spaces; however, the general idea behind this proposed method seems like it could be extended.
> > Limitations And Societal Impact:
> > I think the limitations of the approach were well addressed. One minor limitation I was curious about is how this approach might extend to environments that are dynamic, or action spaces which including manipulating an environment.
>
> This is a good question, and we will add text to the Future Work section (Sec. 7) to address this point. Recent work from Bradley et al (ref [3] in our paper) has shown that the learning over subgoals representation that we leverage for planning under uncertainty can also be extended to support long-horizon multi-stage task planning (for plans specified in Linear Temporal Logic) in partially observable settings. We believe that extending our approach to support this more complex application domain seems a promising direction for future work in this space.

---

### Author Response · Authors · 2021-08-10
**Meta-Rebuttal: highlighting comments and replies shared between reviewers**

We thank all the reviewers for their comments, questions, and suggestions. We feel that the review process has helped us hone the message of our paper as well as highlighted places where we could have been more clear, which we detail in our comments below how we plan to address. We have replied to each reviewer individually---which is where we have included most of the detail during this rebuttal period---and we include here a summary of some of the larger comments and themes:

**An expansion of the discussion of our contributions and a clarification of the limits of our investigation.** A number of the changes we will make in response to the reviewer comments are to clarify the scope of our contributions and the impact of our work, particularly addressing some comments from Reviewer 74uJ. In particular, in addition to our theoretical contributions---which motivate our approach and choice of representation (Sections 1, 2, and 3)---and qualitative results, our quantitative results primarily focus on demonstrating that our explanations are sufficiently rich with information that they can be used to improve the agent's behavior, a key prerequisite if the explanations are ever to be useful in this capacity to a human operator. While we believe that rigorous human trials will be an important component for understanding how to best communicate to human operators, given limited space, we decided to focus on this quantitative investigation of the information content of our explanations and on a clear discussion of our theoretical contributions, which all reviewers highlight as a strength of the paper. In our comments below, we detail how we intend to improve and add text to make these points clearer in the Introduction and in Sections 1, 2, and 3 (where we discuss the representation). In addition, we will expand our discussion in Limitations & Future Work (Sec. 7) to reiterate the importance of further human-in-the-loop trials before our explanations should be used in a mission-critical capacity.

**Improvements to clarity about procedure and expanding on experimental details.** All three reviewers had various suggestions for how to improve the clarity of the paper and requests to expand on the details of some of the experimental procedures. We have enumerated a number of changes throughout the paper to address these, and after making many of these changes locally we believe that the paper will be easier to follow; thank you all for the suggestions! We will clarify details about the problem setting, the information available to the agent, and the planning process. In particular, we emphasize that the agent makes iterative progress towards the goal by alternating between selection of a subgoal via the high-level planner (Sec. 3) and a non-learned low-level motion planner that makes incremental progress towards that subgoal through known space. In addition, we have drafted a number of improvements to the figures, making them slightly larger and adding annotations to make it clearer what the agent plans to do as well as what the symbols represent. We describe these improvements in more depth in our individual replies below.

**Address the form and presentation of our explanations.** As pointed out by Reviewers 74uJ and 5zEg, the presentation of the information content of the explanations, including both the grammar and the accompanying graphical representation, is an important design consideration. These reviewers raised questions about the way to most effectively structure the explanations (as shown in Fig. 1 and Fig. 3) to most effectively communicate the agent's intentions to humans. However, similar to our previous comment, to rigorously evaluate different approaches to presenting our explanations would require significant space and, though important before our explanations are used in a mission-critical capacity, is not the focus of this work. So as to aid in future investigations in this regard, we will add language to Sections 3 (Theory: Explanations) and 7 (Limitations & Future Work) as well as in Appendix A.3 in an effort to elucidate the process by which our grammar was selected as well as suggest alternative ways of presenting our explanations so as to maximize communication.

**Preliminary experiments of pixelwise saliency** Both Reviewers 5zEg and qvY1 asked for an additional discussion of the feasibility of applying existing CNN interpretability techniques to understand how subgoal properties are estimated from raw images. While we feel that rigorous experiments are beyond the scope of this work, we agree with Reviewer 5zEg that a preliminary demonstration of this capability would be helpful for clarity. We have conducted preliminary experiments using an implementation of the "Axiomatic Attribution for Deep Learning" feature attribution technique [26] to interpret predictions of the neural network trained via our procedure (as described in Sec. 4) in the Guided Maze environments. These preliminary results qualitatively agree with our expectations. The attribution for $P_S$ (the likelihood a subgoal will lead to the goal) correctly highlights the green path on the ground as the most salient feature and expectedly assigns positive attribution when the subgoals do lead to the goal and negative attribution when they do not, which we believe supports the potential to enhance our approach via integration with other existing tools. We will create a new Appendix A.4 that includes a figure showing these proof-of-concept results and a brief discussion of their significance.

In the individual reviewer-specific replies below, we include more details of these broader changes as well as address the other (mostly smaller) comments and clarifications.

---

### Decision · Program_Chairs · 2021-09-27

**Decision:**

Accept (Poster)

**Comment:**

The paper proposes a method for generating counterfactual explanations for the decisions of a high-level planner that navigates an agent through a partially known environment. The planner reasons over subgoals and the proposed method performs gradient descent over the parameters of these subgoals to identify changes that cause the planner to choose one subgoal over another. A subsequent round of gradient descent is performed with all but one parameter fixed in an effort to filter out false positives. The results are converted to a textual explanation using a rule-based grammar. The quality of the explanations is evaluated by using them to train the navigational agent following much the same procedure that is used to identify the explanations.

The problem of generating a valid and concise explanation for why a policy/planner chose one action over another is challenging and touches on broader problems in interpretability/explainability that are of interest to many in the Ml community. The reviewers agree that the proposed approach is carefully designed and is technically sound. The method is presented clearly and the paper, as a whole, is well written. The experimental evaluation provides a compelling demonstration of the utility of the resulting explanations. As the reviewers note, while this evaluation is deliberately not subjective, a human evaluation of the explanations would provide more insight into the method's broader utility. However, this can be left for future work.